# Combined Treatment with Acalabrutinib and Rapamycin Inhibits Glioma Stem Cells and Promotes Vascular Normalization by Downregulating BTK/mTOR/VEGF Signaling

**DOI:** 10.3390/ph14090876

**Published:** 2021-08-29

**Authors:** Yu-Kai Su, Oluwaseun Adebayo Bamodu, I-Chang Su, Narpati Wesa Pikatan, Iat-Hang Fong, Wei-Hwa Lee, Chi-Tai Yeh, Hsiao-Yean Chiu, Chien-Min Lin

**Affiliations:** 1Graduate Institute of Clinical Medicine, College of Medicine, Taipei Medical University, Taipei 11031, Taiwan; yukai.su@gmail.com (Y.-K.S.); ichangsu@gmail.com (I.-C.S.); 18149@s.tmu.edu.tw (I.-H.F.); 2Department of Neurology, School of Medicine, College of Medicine, Taipei Medical University, Taipei 11031, Taiwan; 3Division of Neurosurgery, Department of Surgery, Shuang Ho Hospital, Taipei Medical University, New Taipei City 23561, Taiwan; 4Taipei Neuroscience Institute, Taipei Medical University, Taipei 11031, Taiwan; 5Department of Medical Research & Education, Shuang Ho Hospital, Taipei Medical University, New Taipei City 23561, Taiwan; dr_bamodu@yahoo.com (O.A.B.); narpatisesa@gmail.com (N.W.P.); ctyeh@s.tmu.edu.tw (C.-T.Y.); 6Doctorate Program of Medical and Health Science, Faculty of Medicine, Public Health, and Nursing, Universitas Gadjah Mada, Yogyakarta 55281, Indonesia; 7Department of Pathology, Shuang Ho Hospital, Taipei Medical University, New Taipei City 23561, Taiwan; whlpath97616@s.tmu.edu.tw; 8Department of Medical Laboratory Science and Biotechnology, Yuanpei University of Medical Technology, Hsinchu 300, Taiwan; 9College of Nursing, Taipei Medical University, Taipei City 11031, Taiwan

**Keywords:** glioblastoma, BTK, mTOR, cancer stem cells, vascular normalization

## Abstract

Glioblastoma (GBM) is the most common primary malignant brain tumor in adults, with a median duration of survival of approximately 14 months after diagnosis. High resistance to chemotherapy remains a major problem. Previously, BTK has been shown to be involved in the intracellular signal transduction including Akt/mTOR signaling and be critical for tumorigenesis. Thus, we aim to evaluate the effect of BTK and mTOR inhibition in GBM. We evaluated the viability of GBM cell lines after treatment with acalabrutinib and/or rapamycin through a SRB staining assay. We then evaluated the effect of both drugs on GBM stem cell-like phenotypes through various in vitro assay. Furthermore, we incubated HUVEC cells with tumorsphere conditioned media and observed their angiogenesis potential, with or without treatment. Finally, we conducted an in vivo study to confirm our in vitro findings and analyzed the effect of this combination on xenograft mice models. Drug combination assay demonstrated a synergistic relationship between acalabrutinib and rapamycin. CSCs phenotypes, including tumorsphere and colony formation with the associated expression of markers of pluripotency are inhibited by either acalabrutinib or rapamycin singly and these effects are enhanced upon combining acalabrutinib and rapamycin. We showed that the angiogenesis capabilities of HUVEC cells are significantly reduced after treatment with acalabrutinib and/or rapamycin. Xenograft tumors treated with both drugs showed significant volume reduction with minimal toxicity. Samples taken from the combined treatment group demonstrated an increased Desmin/CD31 and col IV/vessel ratio, suggesting an increased rate of vascular normalization. Our results demonstrate that BTK-mTOR inhibition disrupts the population of GBM-CSCs and contributes to normalizing GBM vascularization and thus, may serve as a basis for developing therapeutic strategies for chemoresistant/radioresistant GBM.

## 1. Introduction

Glioblastoma (GBM) is the most common primary malignant brain tumor in adults, with a median duration of survival of approximately 14 months after diagnosis [1]. As one of the standard therapeutic approaches, chemotherapy is effective for reducing the tumor size, inhibiting distant metastasis, and prolonging patient survival. However, GBM exhibits a high resistance to chemotherapy, and recurrence is commonly observed [2]. Emerging evidence indicates a strong link between resistance to chemotherapy and the induction of epithelial–mesenchymal transition (EMT) in cancer [3]. EMT is a process during which cells undergo morphologic changes from the epithelial phenotype to mesenchymal phenotype, resulting in enhanced motility and increased invasion, proliferation, and resistance to apoptosis [4,5]. Among the heterogeneous cell populations within GBMs, one cell subset, termed glioma cancer stem cells (GCSCs), reportedly exhibits stem cell properties, such as an intrinsic long-term ability to self-renew, the capability to differentiate into progeny that is non-tumorigenic but still contributes to tumor growth, and the ability to drive the progression of GBM by enhancing resistance to therapy and/or post-response re-population of GBM cells [6]. These cells acquire the aforementioned properties during malignant progression by reactivating a complex process called EMT, which is integral in embryonic development, wound healing, and CSCs behavior [7]. CSCs demonstrate a robust capacity for self-renewal and play a prominent role in tumor initiation. In addition, they exhibit a strong ability to undergo EMT, which contributes to their high metastatic potential. Notably, CSCs are also resistant to radiotherapy and several currently available chemotherapy regimens [8]. The maintenance of CSCs is mediated by a core set of key transcription factors, which are under the control of a large range of developmental signals and extracellular cues [9]. Among these cues, Bruton’s tyrosine kinase (BTK) has strong potential as an oncogenic driver.

BTK is a nonreceptor cytoplasmic tyrosine kinase that is primarily expressed in cells of hematopoietic lineage. It is one of the five members of the Tec family of nonreceptor tyrosine kinases, which consists of Tec, BTK, interleukin-2-inducible T cell kinase (ITK), bone marrow tyrosine kinase in chromosome X (BMX), and tyrosine-protein kinase/resting lymphocyte kinase (TXK/RLK). Mutations in the BTK gene result in X chromosome–linked agammaglobulinemia in humans and X chromosome–linked immunodeficiency in mice [10]. This suggests that BTK plays an important role in the growth and differentiation of blood cells. Recent studies have shown that BTK is an important mediator in coupling activated immunoreceptors to downstream signaling events that affect diverse biological functions, from cellular proliferation, differentiation, and adhesion to innate and adaptive immune responses [11,12,13]. Furthermore, BTK is actively involved in the intracellular signal transduction of G-protein–coupled receptors, lymphocyte surface antigens, cytokine receptors, toll-like receptors, and integrin molecules, the cross-talk of which plays a critical role in the tumorigenesis of several malignancies [14,15].

More evidence on the role of BTK signaling in solid cancers has begun to emerge. In 2016, Gunderson et al. reported the use of ibrutinib, a first-generation BTK inhibitor, for treating pancreatic cancer [16]. They demonstrated that the pharmacological inhibition of BTK can activate adaptive immune responses in patients with pancreatic cancer, presenting a potential therapeutic regimen for treating this type of cancer. In multiple myeloma, an elevated expression of BTK leads to Akt/Wnt/BTK-dependent upregulation of key stemness genes (OCT4, SOX2, NANOG, and MYC) and enhanced self-renewal [17]. Enforced expression of BTK in this cancer enhances the features of cancer stemness and its resistance to conservative myeloma chemotherapy [17]. In a recent publication, we revealed the oncogenic role of BTK in ovarian cancer and GBM. Using molecular knockdown and ibrutinib, we demonstrated that the inhibition of BTK signaling markedly reduces the subpopulation of CSCs and enhances the anticancer effects of conservative chemotherapy in ovarian and GBM cancer cells [18,19]. Notably, BTK was transiently upregulated in vivo in endothelial cells of the ovary, and its mRNA was continually induced in human umbilical vein endothelial cells (HUVECs) [20]. Following this evidence, a study demonstrated that the delivery of ibrutinib by sialic acid–stearic acid conjugated therapy into tumor-associated macrophages reduced angiogenesis and suppressed tumor growth [21]. In addition, an in vitro screening revealed a synergistic combination of the mTOR kinase inhibitor everolimus and BTK inhibitor PLS-123. This combination proved to be effective in inducing apoptosis in mantle cell lymphoma and repressing JAK2/STAT3 and Akt/mTOR signaling pathways and SGK1 expression, accentuating the use of this therapeutic modality as a potential strategy in treating cancers [22].

Based on this extensive evidence and the findings of our previous work on BTK and its potential synergism with mTOR inhibition, we hypothesized that targeting this particular molecule would effectively suppress the CSCs-like phenotype of GBM and, through combination therapy, ultimately reduce angiogenesis and improve the vascular normalization of GBM tumors. In this study, we aimed to substantiate the combined inhibition of BTK and mTOR signaling as an appealing drug target and alternative targeted therapy for GBM.

## 2. Materials and Methods

### 2.1. Reagents and Drugs

Acalabrutinib (ACP-196, ≥99% (HPLC), Cat. No. S8116) and rapamycin (AY-22989, ≥99% (HPLC), Cat. No. S1039) were purchased from Selleck Chemicals (Houston, TX, USA). They were dissolved at ~0.5 mg/mL in a 1:1 solution of dimethyl sulfoxide (DMSO): phosphate-buffered saline (PBS) at pH 7.2 to prepared stocks of 10 mM and stored at −20 °C until use. The stock solution was further diluted in a sterile culture medium immediately prior to use. 

### 2.2. Cells and Cell Cultures

Human GBM cell lines, namely U87MG (ATCC HTB-14) and LN229 (ATCC^®^ CRL-2611™), were purchased from American Type Culture Collection (ATCC; Manassas, VA, USA). The cells were cultured in Dulbecco’s modified Eagle’s medium (#12491023; GIBCO, Life Technologies Corp., Carlsbad, CA, USA) and were supplemented with 10% fetal bovine serum (GIBCO, Life Technologies Corp.), penicillin (100 IU/mL), and streptomycin (100 µg/mL) (#15140122, GIBCO, Life Technologies Corp.) in an incubator with 5% CO_2_ in humidified air. The cells were sub-cultured, and media changed when >97% confluence was attained. All cells used in the study were ≤passage 4. Cell identity by the vendor using short tandem repeat analysis, and, cell purity based on absence of adventitious cellular or microbial contaminants and potential cross-contaminations with other cell lines was confirmed by the vendor, ATCC.

### 2.3. Western Blotting

After trypsinization, GBM cells were harvested and lysed. Protein lysates were then heated, and immunoblotting was performed with 5% skimmed milk in Tris-buffered saline with Tween 20 (TBST) for 1 h and incubated overnight at 4 °C with primary antibodies (Appendix A) against the protein of interest. PVDF membranes were washed several times with TBST after incubation with the primary antibody and then incubated with a horseradish peroxidase-labeled secondary antibody at room temperature for 1 h and rewashed with TBST. Band detection was subsequently performed using enhanced chemiluminescence, western blotting reagents, and a BioSpectrum Imaging System (UVP; Upland, CA, USA).

### 2.4. Sulforhodamine B Cytotoxicity Assay

Drug cytotoxicity and cell viability were assessed using a sulforhodamine B (SRB) assay, as previously described [23]. First, aliquots of 3 × 10^3^ GBM cells were seeded in 96-well microtiter plates, with each well containing the supplemented medium, and incubated in humidified 5% CO_2_ in air at 37 °C for 24 h. Thereafter, some cells were treated with various concentrations of acalabrutinib or rapamycin, and untreated cells served as controls. Quantification was performed twice in triplicate experiments. Optical density was measured at a wavelength of 495 nm by using a SpectraMax microplate reader (Molecular Devices, Kim Forest Enterprises Co., Ltd., Taipei, Taiwan).

### 2.5. Colony-Formation Assay

First, 0.15 × 10^3^ U87MG and LN229 cells per well were cultured in 6-well plates. After 24-h incubation, cells were treated with or without acalabrutinib, rapamycin, or both. The medium was replaced every 72 h with a fresh medium. The media were discarded after treatment for 12 days, and the cell colonies that formed were stained with 0.1% crystal violet in 20% methanol and counted. Images were obtained using a microscope and a UVP BioSpectrum Imaging System.

### 2.6. Tumorsphere-Formation Assay

U87MG and LN229 cells were pretreated with or without acalabrutinib, rapamycin, or both. They were then seeded at a density of 800 cells/mL of tumorsphere-formation medium in ultralow attachment 12-well plates for 12 days. The medium comprised HEScGRO serum-free medium (Chemicon, Cat. SCM020) with 20 ng/mL of hEGF (NeuroCult NS-A basal medium) and 10 ng/mL of hFGF-b (STEMCELL Technologies, Vancouver, Canada; Cat. 5751). The formed tumorspheres (spherical, nonadherent cell masses with a diameter > 90 μm) were photographed and counted under an inverted phase contrast microscope.

### 2.7. VEGF Enzyme-Linked Immunosorbent Assay (ELISA)

In total, 200 µL of conditioned media were collected from samples in triplicate and then processed using Quantikine Human VEGF ELISA Kits (R&D Systems, Minneapolis, MN, USA) according to the directions of the manufacturer. 

### 2.8. Endothelial Migration Assays

Media conditioned by parental or GBM sphere populations for 24 h were added to the bottom chambers of 24-well tissue culture plates in triplicate. Human microvascular endothelial cells (20,000) were added to the upper chambers of transwell assays (BD Biosciences, Franklin Lakes, NJ, USA). In experiments with drug supplementation, conditioned media from parental or GBM sphere populations that had been treated with acalabrutinib and/or rapamycin were added to the upper chambers of 24-well tissue culture plates in triplicate. Cells were allowed to invade for 14 h before being fixed, stained, and quantified.

### 2.9. Endothelial Tube-Formation Assays

Standard HC BD Matrigel Matrix (Cat. No. 354248, BD Biosciences) was diluted with media conditioned for 24 h by parental or GBM sphere populations and coated in 24-well culture plates in triplicate at 37 °C for 1 h. Human microvascular endothelial cells (20,000) were added to the transwell chambers (BD Biosciences). In experiments with drug supplementation, conditioned media from parental or GBM sphere populations previously treated with acalabrutinib, rapamycin, or both were added to the chambers of 24-well tissue culture plates in triplicate. Cells were incubated for 16 h, imaged, and quantified. The tube formation ability of human microvascular endothelial cells was measured at indicated time points.

### 2.10. Histology and Immunostaining

Brains with tumors were collected from U87MG-bearing (2 × 10^6^ U87MG cells inoculated in the hind flanks) 4–6-week old female NOD/SCID mice purchased from BioLASCO (BioLASCO Taiwan Co., Ltd., Taipei, Taiwan) (median weight, 17.4 ± 2.1 g) on Day 21, at the end of treatment. The mice were randomly placed into the control/vehicle (*n* = 10), acalabrutinib (*n* = 10), rapamycin (*n* = 10), or acalabrutinib–rapamycin combination (*n* = 10) group. Intraperitoneal administration of acalabrutinib (20 mg/kg), rapamycin (8 mg/kg), or combination therapy was initiated on Day 6, when the tumor started to become palpable and continued every 48 h over the next 12 days. Tumor size was measured on days 6, 9, 12, 15, 18, and 21 after GBM cell inoculation by using calipers, and the tumor volume (v) was calculated using the standard formula: v = length (l) × width (w)^2^ × 0.5. At the end of the experiment (Day 21), the tumor-bearing mice were humanely sacrificed, and the tumors were extracted, examined, photographed, and measured again. Necrotic areas were excluded from analysis. Tumor microvascular density (MVD), perivascular cell coverage of vessels, coverage of the basement membrane (BM), proliferation, apoptosis, and macrophage density were assessed using a semiautomatic in-house MATLAB (MathWorks) segmentation algorithm. The thickness of the BM was assessed using ImageJ. The quick (Q)-score used for quantification of immunohistochemical staining using the formula Q = P × I, where P is the percentage of positively stained cells and I is the intensity of staining. The maximum Q-score was 300. The animal study was approved by the Institutional Laboratory Animal Committee of the Taipei Medical University (Approval number: LAC-2020-0550).

Primary antibodies included CD31 (endothelial cells; 1:200; MAB1398Z; EMD Millipore), desmin (perivascular cells; 1:500; AF3844; R&D Systems), collagen IV (BM; 1:2000; AB7569; EMD Millipore), and Alexa Fluor 647-conjugated F4/80 (macrophages; 1:50; MCA497A647; AbD Serotec).

### 2.11. Statistical Analysis

All experiments were performed twice in triplicate, and the data are presented as the mean ± standard error of the mean. All statistical analyses were performed using Student’s *t* test on GraphPad Prism 5 software (GraphPad Software Inc., La Jolla, CA, USA). A result of *p* < 0.05 was considered statistically significant.

## 3. Results

### 3.1. Rapamycin and Acalabrutinib Effectively Reduced the Viability of GBM Cell Lines and Exerted a Synergistic Antiproliferation Effect

Previous studies have shown the effectiveness of rapamycin in inhibiting the tumorigenesis of GBM cells. Here, we aimed to corroborate these findings and establish a basis for our working concentration in subsequent experiments. We treated GBM cell lines with various doses of rapamycin ranging from 0.01 μM up to 1 μM, with IC_50_ values of 0.1 μM and 0.125 μM (U87MG and LN299, respectively) (Figure 1A). Next, we used a novel, selective BTK inhibitor (acalabrutinib) and demonstrated its antiproliferative effect on GBM cells. Using two GBM cell lines used in a previous experiment with rapamycin, we examined the effects of acalabrutinib through SRB assay. Cells were treated with acalabrutinib at concentrations ranging between 2.5 μM and 10 μM for 48 h. This treatment markedly reduced the viability of U87MG and LN299 cells, with IC_50_ values of at least 11 μM and 12 μM, respectively (Figure 1B). We hypothesized that BTK inhibition would enhance the effect of mTOR inhibition on GBM cell lines. Thus, we incubated U87MG and LN299 simultaneously with rapamycin and acalabrutinib for 48 h and evaluated their effect through SRB assay. Cells exposed to rapamycin combined with acalabrutinib exhibited a considerably enhanced antiproliferative effect of rapamycin and considerably reduced viability of GBM cell lines (Figure 1C,D). The Chou–Talalay algorithm-based CompuSyn software was used for isobologram-aided combinatorial analysis of rapamycin with acalabrutinib, and the results demonstrated that all of the combination points within the right-angled isobologram triangle had combination index scores of <1 for all corresponding dose combinations, indicating synergism between the two agents. Combined ibrutinib and torin2 also exhibits synergism on the viability of U87MG and LN229 (Appendix A). These results highlight the potential role of therapeutically targeting BTK and mTOR when treating patients with GBM.

### 3.2. Combination Therapy of Acalabrutinib and Rapamycin Considerably Reduced the Tumorsphere-Formation Potential of GBM Cells 

We evaluated the effects of pharmacologically blocking both BTK and mTOR pathway signaling on the maintenance of the CSCs properties of GBM cells. First, we examined the effects of 5 µM acalabrutinib and 0.1 µM rapamycin on U87MG and LN229 cell lines. The results of this experiment revealed that a single administration of each drug considerably reduced colony numbers (75% and 50% of control, respectively), and simultaneous incubation with these two drugs markedly enhanced their inhibitory effects (up to 25% compared with the control; Figure 2A). Next, we demonstrated that a small combination dose of 5 µM acalabrutinib and 0.1 µM rapamycin considerably reduced both the size and number of spheres formed by U87MG and LN229 cells by up to 49.5% and 51%, respectively (Figure 2B). Furthermore, western blot analysis of the formed GBM spheres confirmed reduction in expression of CSCs markers Nestin, SOX2, OCT4, CD133, KLF4, and NANOG but increased the expression of differentiated neuron markers GFAP and NEFL. The combination of acalabrutinib and rapamycin further enhanced or inhibited the effect of each drug on these markers (Figure 2C). These results indicated that the pharmacologic inhibition of BTK and mTOR signaling effectively abolished the CSCs-like properties of GBMs.

### 3.3. GBM CSCs Consistently Secreted Markedly Elevated Levels of Vascular Endothelial Growth Factor

The effect of GBM CSCs on angiogenesis has been previously described [24]. To validate this, we performed a similar experiment using Quantikine human vascular endothelial growth factor (VEGF) ELISA kits and observed that GBM tumorspheres secreted higher levels of VEGF than did their parental cells. Hypoxia further increased the expression of VEGF in the GBM tumorsphere population (Figure 3A,B). This validates previous findings that GBM CSCs have a higher expression of VEGF, especially in hypoxic conditions. In previous experiments, we demonstrated the effects of pharmacologically inhibiting the BTK and mTOR pathways on GBM CSCs. Accordingly, we subjected the aforementioned GBM tumorsphere cells to 5 µM acalabrutinib, 0.1 µM rapamycin, or a combination of both after exposing them to hypoxia and then evaluated their VEGF expression. As expected, VEGF expression was markedly diminished after treatment with acalabrutinib or rapamycin, and this inhibition was further enhanced when both drugs were co-administered (Figure 3C). These results support the notion that BTK and mTOR inhibition effectively attenuate VEGF expression by GBM CSCs. 

### 3.4. Targeting VEGF Inhibited the Effects of GBM CSCs on Endothelial Cells

To assess the effect of the combined inhibition of BTK and mTOR on endothelial cells, we adopted an in vitro model of angiogenesis with the HUVEC cell line cocultured on matrigel with parental or sphere-conditioned media. We discovered that HUVEC grown with sphere-conditioned media exhibited increased migration (Figure 4A) and tube formation (Figure 4B; also see Table 1). These effects were markedly diminished when the sphere cells of U87MG were preincubated with acalabrutinib or rapamycin over the course of 48 h. The effects were further enhanced by the simultaneous administration of both drugs on the cells (Figure 4A,B, also see Table 1). We confirmed that the pharmacological inhibition of BTK and mTOR further downregulated the expressions of VEGF and ANGPT2, upregulated the expression of ANGPT1, resulting in increased ANGPT1/ANGPT2 ratio in the U87MG and LN229 cell lines (Figure 4C,D). These results demonstrate that inhibiting VEGF expression by reducing the number of GBM tumorspheres is a valid approach to inhibiting neoangiogenesis and enhancing vascular normalization in GBM tumors. 

### 3.5. Acalabrutinib Inhibited BTK, Akt, and ERK Signaling, Enhancing Rapamycin Inhibition of mTOR Activation

We postulated that BTK inhibition may synergize with rapamycin through its interaction with the Akt signaling pathway, an upstream signaling axis of mTOR, because they have previously been shown to interact in B-cells [25]. U87MG and LN229 cells were administered 5 µM acalabrutinib, 0.1 µM rapamycin, or both for 48 h. Target protein expression was analyzed through western blotting. Our results demonstrated that as expected, acalabrutinib inhibited the phosphorylation of BTK (p-BTK) and AKT (p-AKT) proteins, while rapamycin treatment had no apparent effect on the expression of Akt signaling and was only effective in inhibiting mTOR activation. However, the co-administration of acalabrutinib and rapamycin significantly suppressed the expression of p-BTK, p-AKT, p-mTOR, p-S6K, and S6K proteins (Figure 5A,B). This experiment revealed that acalabrutinib synergizes with rapamycin through the inhibition of BTK/Akt signaling axis, at least at the protein expression level. 

### 3.6. Combination Treatment with BTK Inhibitor Acalabrutinib and mTOR Inhibitor Rapamycin Reduced In Vivo Tumorigenesis and Angiogenesis through the Inhibition of the BTK/Akt/mTOR Signaling Axis

We demonstrated that BTK signaling plays a critical role in maintaining the CSCs-like properties and viability of GBM in vitro. Hence, we sought to replicate our in vitro findings through an in vivo approach. After inoculating female NOD/SCID mice with 2 × 10^6^ U87MG cells, the tumors became palpable on Day 6, and the tumor-bearing mice were randomly assigned to the control/vehicle, acalabrutinib, rapamycin, or combination group. Intraperitoneal administration of 20 mg/kg acalabrutinib significantly suppressed tumor growth; compared with the vehicle-treated group, acalabrutinib-treated mice exhibited a 1.43-fold (*p* < 0.01) reduction in tumor size. Similarly, rapamycin-treated mice exhibited a 1.93-fold (*p* < 0.01) reduction in tumor size. Notably, compared with the vehicle group, the group of mice treated with a combination of acalabrutinib and rapamycin showed the most significant reduction in tumor size, with a 2.50-fold reduction (*p* < 0.01) (Figure 6A). Moreover, because no significant weight loss was observed in any of the groups, these treatments also demonstrated that the drugs have relatively low toxicity (Figure 6B). Furthermore, ex vivo TUNEL assay of tumor samples extracted from the mice revealed that, compared with cells from the control mice, those from the drug-treated tumor-bearing mice had a higher intensity of staining. Mice treated with combination therapy had the highest count of TUNEL-positive cells among all of the groups, suggesting a higher rate of apoptosis in this group (Figure 6C). Tissues extracted from the mice were subjected to IHC analysis, the results revealed that, compared to the control, treatment with rapamycin caused mild-moderate downregulation of p-BTK, p-AKT, p-S6K, and VEGF protein expression levels, acalabrutinib strongly suppressed the expression of these proteins, and co-administration of acalabrutinib and rapamycin markedly suppressed the expression levels of p-BTK, p-AKT, p-S6K, and VEGF proteins, further indicating synergism (Figure 6D). To determine the effects of a combination therapy of acalabrutinib and rapamycin on tumor vasculature, the morphology of vessels extracted from mice tumor samples was assessed (Figure 6E,F). Tissue staining showed increased tumor MVD in single-drug treatment groups, with the group treated with combination therapy exhibiting the greatest increase in MVD (Figure 6G). Perivascular cell coverage (represented as % Desmin/CD31 ratio) was higher in the combination therapy group than in the acalabrutinib or rapamycin monotherapy groups (Figure 6H). Similarly, considerably increased coverage of BM was observed in the combination therapy group compared with the monotherapy group (Figure 6I). In summary, these data at least partially indicate that acalabrutinib–rapamycin combination therapy impairs tumorigenesis through the suppression of GBM stemness signaling and negative modulation of the BTK/Akt/mTOR signaling pathway, ultimately affecting angiogenesis and vascular normalization in vitro and in vivo and that this combination therapy can be used as a therapeutic strategy with promising efficacy for patients with GBM.

## 4. Discussion

BTK is a nonreceptor cytoplasmic tyrosine kinase that acts as a cellular sensor and transducer of physical and chemical stress. Increasing evidence has revealed the pleiotropic roles of BTK in cellular and physiological responses in different cell types; however, their precise physiological roles in GBM remain unclear. Our study further confirms that BTK maintains CSCs-like properties in GBM. Using a pharmacological approach, we demonstrated that acalabrutinib inhibits cell proliferation, markedly reduces metastatic potential, and attenuates the tumorsphere-forming capacity of GBM cells. Combined with mammalian target of rapamycin (mTOR) inhibition, acalabrutinib further reduces tumorsphere formation and considerably reverts vascular normalization markers in GBM cells.

BTK has been demonstrated to be a central player of diverse modulatory roles in many cellular processes. Some studies have reported the important role of BTK signaling in cell proliferation. Ibrutinib considerably reduces cell proliferation in cases of ovarian cancer. It targets epidermal growth factor receptor (EGFR) mutation in hepatocellular carcinoma (HCC), effectively reducing HCC cell proliferation and sensitizing the tumor cells to sorafenib treatment [26]. Ibrutinib also targets ERBB family kinases signaling in HER2 breast cancer cells, effectively reducing their proliferation [27]. In line with these findings, we previously discovered that ibrutinib inhibits the proliferation of GBM cells [18]. We obtained similar results by using a more specific second-generation BTK inhibitor, acalabrutinib, confirming that BTK inhibition effectively reduces cell proliferation in GBM. 

CSCs have been proposed to be one of the drivers of cancer aggressiveness. Because of their stem cell’s properties, these cells can perpetuate themselves through autorestoration [28]. CSCs express ABC transporters and may export chemotherapy drugs, giving cancers their drug resistance capabilities. In addition, CSCs are more resistant to apoptosis. Colon cancer CSCs overexpress LGR5, which results in increased numbers of antiapoptotic Bcl-2 and BcL-xL genes [29]. Chikamatsu et al. showed that CD44+ CSCs of head and neck squamous cell carcinoma are more resistant than CD44− CSCs to apoptosis [30]. These traits of CSCs, along with their increased potential to migrate to other organs and initiate metastasis, prompted many researchers to attempt the elimination of this subpopulation of cells [31]. Our experiments provide evidence that BTK maintains the population of CSCs in GBM. A low dose of acalabrutinib effectively inhibited the expressions of BTK and key stem cell marker proteins. Phenotypically, our results indicate that treatment with acalabrutinib markedly diminishes the sphere-formation and colony-formation abilities of GBM cells. These results corroborate the findings of other studies that BTK plays an important role in maintaining the population of CSCs. Zucha et al. found that the chemoresistance of ovarian cancer cells is dependent on Btk and JAK2/STAT3, which maintain CSCs by inducing the expression of Sox-2 and prosurvival genes [19]. Our previous study on glioma revealed that BTK knockdown considerably attenuates tumorsphere formation and enhances the anticancer effects of temozolomide on GBM cells [18].

GBM CSCs are increasingly implicated in the dismal prognosis of GBM, with mortality within 12 to 24 months of diagnosis. This has been associated with the inherent capacity of the CSCs to elicit and drive radio- and chemotherapy resistance in GBM cells, and the fact that these CSCs are maintained in vivo in a niche that is characteristically hypoxic (reduced oxygen tension) [32]. It is therefore clinically relevant that we found that GBM tumorspheres secreted higher levels of VEGF than did their parental counterparts, and that hypoxia further increased the expression of VEGF in the GBM tumorsphere population. This finding is consistent with reports that hypoxia promotes vessel growth by enhancing various pro-angiogenic pathways that mediate vital components of endothelial, stromal, and vascular support cell biology [33]. “Interestingly, recent studies show that hypoxia influences additional aspects of angiogenesis, including vessel patterning, maturation, and function” [33]. This is also in line with our current findings and should inform the design and development of any efficacious anti-GBM treatment modality.

Anti-angiogenic modalities are recognized as viable strategies for treating malignant tumors [34]. Studies have reported the effectiveness of agents such as bevacizumab (Avastin) as a component of a combination therapy regimen in some malignancies, including colon cancer and GBM [34,35]. Several theories have been proposed to clarify the clinical mechanism of antiangiogenic drugs. Some studies have revealed that some cancers express VEGF receptors themselves, thus anti-VEGF may give them direct antitumor effects [36]. Moreover, anti-angiogenic treatments have been shown to disrupt the vascular structure of tumors, depriving them of nutrition and promoting severe hypoxia [37,38]. This mechanism, however, may also contribute to the perceived ineffectiveness of anti-angiogenic agents in treating GBM, as the disrupted vessels may restrict drug delivery, decreasing their clinical efficacy [32]. In light of this issue, researchers began to focus on another possible mechanism of the anti-angiogenic agent which is that antiangiogenics may “normalize” tumor vasculature, improving its efficiency in delivering chemotherapeutic drugs and other therapeutic modalities [39,40]. Several studies have shown that vascular normalization may have a beneficial effect in tumors. Park et al. found that the combination of Tie2 activation and Ang2 inhibition is a potent therapeutic approach for eliciting a tumor microenvironment that favors improved chemotherapeutic drug delivery into tumors [41]. Recently, it was shown that, compared to VEGF-targeting alone, blocking VEGF, Ang-2, and programmed cell death protein-1 (PD-1) dramatically increased the survival of syngenic orthotopic GBM mice models [42]. Having said this, we do note that in a systematic review and meta-analysis of randomized clinical trials, Xiao et al. concluded that while anti-VEGF agents do not improve the overall survival (OS) of patients with GBM, they do improve the progression-free survival (PFS) of these patients [43].

Furthermore, consistent with the findings of the present study, several studies have discussed the contribution of CSCs to tumor angiogenesis. Folkins et al. found that compared with non-CSCs, CSCs produce much higher levels of VEGF in both normoxic and hypoxic conditions and that this CSCs-mediated VEGF production leads to amplified endothelial cell migration and tube formation in vitro. When this endothelial migration and tube-formation assays were supplemented with the VEGF-blocking antibody bevacizumab, the in vitro endothelial cell behaviors were blocked. Moreover, in vivo administration of bevacizumab strongly inhibited the growth, vascularity, and hemorrhage of xenografts derived from CSCs, whereas no effects were observed on xenografts from non-CSCs [42]. A VEGF-overexpression glioma model has recently provided supportive evidence for this finding by demonstrating that GBM CSCs overexpressing VEGF produce larger, more vascular, and highly hemorrhagic tumors [44]. This enrichment with CSC-like cells has also been implicated in the resistance of GBM cells to anti-VEGF therapy alluded above [44]. Thus, the elimination of CSCs is a potential therapeutic target for improving vascular normalization in cancers.

To the best of our knowledge, this is the first study to demonstrate that BTK inhibition attenuates the angiogenesis of GBM cells and enhances vascular normalization through the elimination of GBM CSCs. Cells treated with acalabrutinib exhibited a lower VEGF expression and an improved ANPGT1/ANPGT2 ratio. This suggests that BTK has an important role in the angiogenesis of GBM cells. Previously, everolimus, an mTOR inhibitor, was shown to synergize with ibrutinib with respect to its anticancer effects in vitro and in vivo, with BTK-mTOR inhibition leading to the inhibition of the Akt/mTOR and JAK2/STAT3 signaling pathways [22]. We sought to broaden the scope of knowledge regarding the effect of this combination on the angiogenesis of GBM. Indeed, BTK signaling inhibition alone reduces the expression of proangiogenic proteins; supplementation with an mTOR inhibitor causes further reductions and improves the expression of antiangiogenic proteins in GBM cells. Consequently, this combination also further improves the ANGPT1/ANGPT2 ratio. This finding is of clinical significance in the light of the divergent roles of the angiopoietins; with an upregulated expression of ANGPT2 associated with pathological vascular remodeling, and implicated in vascular destabilization, thus, promoting angiogenesis, tumor growth and metastasis [45]. Conversely, ANGPT1 has been shown to quell angiogenic signals and stabilize vessels after disruptive angiogenic processes [46]. More so, there is evidence of the prognostic significance of the ANGPT1/ANGPT2 ratio patients with primary GBM [46]. In addition, targeting ANGPT2 has been shown to elicit significant suppression of angiogenesis, normalize tumor-associated vessels, and inhibit metastasis [47]. Thus, further highlighting the potential role of BTK/mTOR inhibition as an effective antiangiogenic therapeutic strategy in patients with GBM.

The efficacy of any anti-GBM therapeutic measure would be largely dependent on the integrity of the blood-brain barrier (BBB). The distribution of any drug across the BBB is often limited by the physico-biochemical barriers, including tight junctions, which are known to prevent paracellular transport of drugs, and efflux transporters which actively reduce the penetration and intracerebral distribution of any drug. Thus, in terms of pharmacokinetics and pharmacodynamics, earlier reports indicate that acalabrutinib exhibits a 101 ss/F volume of distribution (Vd) to plasma, has a maximum serum concentration (C_max_) of 323 ng/mL, an area under curve (AUC) of 1111 ng h/mL, reaches C_max_ in 0.75 h (T_max_), an elimination half-life (t1/2) of 0.9 h, and a clearance rate of 159 L/h [46]. Consistent with this, a recent report has shown that acalabrutinib may cross the blood-brain barrier [48], while several other clinical trials to evaluate the efficacy of acalabrutinib in treating CNS tumor are in progress [49,50]. Moreover, a recent study has also demonstrated that ibrutinib elicits temporary disruption of the brain endothelial monolayer by inducing cytoskeletal instability and inhibiting ABC transporter [51].

Among the known inhibitors of BTK, acalabrutinib has the least off-target rate and greatest selectivity, followed by zanubrutinib and ibrutinib [46]. Compared to ibrutinib, acalabrutinib has a shorter half-life, is administered once a day in the clinic, and exhibits a higher BTK occupancy with increased dosage. The good balance between fast absorption and fast elimination of acalabrutinib facilitates the rapid inhibition of BTK, which reduces the risk of off-target issues or interaction with other drugs [46]. Because of its relatively short half-life and selectivity, acalabrutinib is able to elicit absolute and perpetual inhibition of BTK without the toxicities associated with off-target inhibition of alternative kinases [47].

Importance of Study: GBM is the most common and lethal brain tumor with a median survival of approximately 14 months after diagnosis. Chemoresistance remains one of the main problems that urgently need to be addressed regarding this tumor. Vascular normalization recently emerged as a highly potential strategy against cancer drug resistance. BTK and Akt/mTOR signaling have been previously shown to maintain CSCs population and drive a tumor vascularization response. Thus, targeting these small molecules may serve as a potential therapeutic strategy to overcome GBM chemoresistance. Our preclinical studies identified that pharmacological inhibition of both BTK and mTOR reduced the GBM cell lines CSCs’ maintenance capabilities and greatly affected HUVEC cells migration and tube formation abilities. These findings were corroborated in our in vivo study. Collectively, these results indicate that BTK/mTOR inhibition improves tumor response and may serve as the basis for clinical trials testing acalabrutinib/rapamycin combination therapy for patients with high-grade gliomas.

In summary, our results, as shown in Figure 7, demonstrate that BTK-mTOR inhibition disrupts the population of GBM CSCs and contributes to normalizing GBM vascularization, which in turn facilitates the delivery of anticancer drugs and reduces the hypoxia of tumor cells. The findings of this study can serve as the basis for future experiments and studies aiming to normalize tumor cell vasculature through the eradication of CSCs and can promote the development of therapeutic strategies based on the inhibition of BTK-mTOR signaling to improve the treatment of chemoresistant/radioresistant GBM. Key points of this study. 1. The combination of the BTK inhibitor, acalabrutinib, and mTOR inhibitor, rapamycin, synergistically and effectively reduced GBM cell viability and negatively regulated GBM-CSCs. 2. The combination of acalabrutinib and rapamycin inhibits angiogenesis and increases vascular normalization through disruption of the BTK/mTOR/VEGF signaling axis.

## Figures and Tables

**Figure 1 pharmaceuticals-14-00876-f001:**
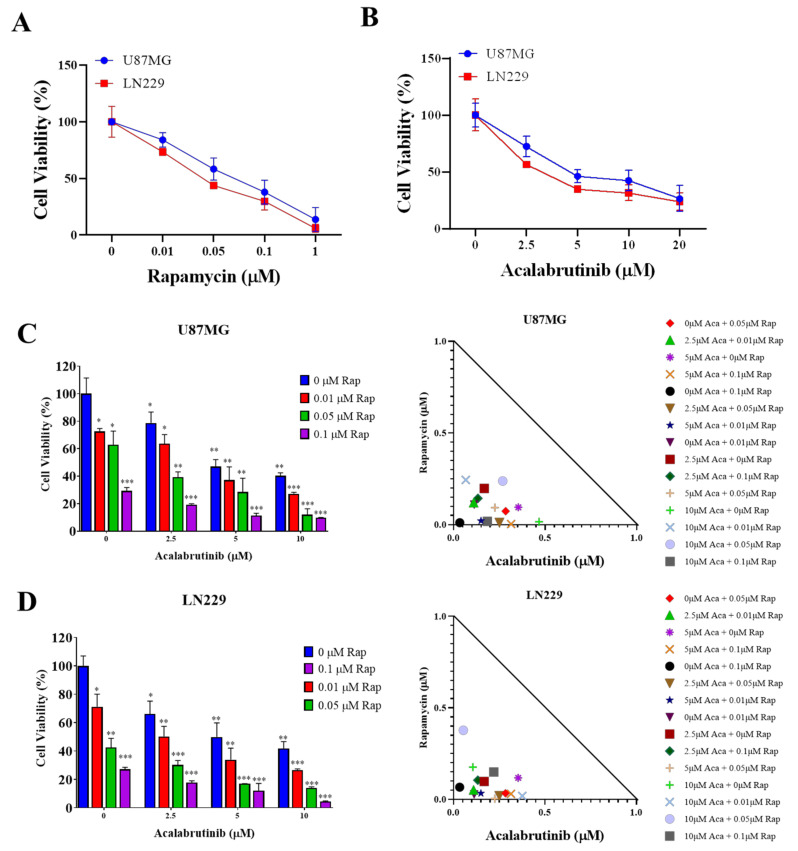
Combined acalabrutinib and rapamycin exhibits synergism. SRB assay for 48 h revealed the effectiveness of (**A**) rapamycin and (**B**) acalabrutinib in inhibiting the viability of GBM cell lines. Graphical representation and isobolograms of the effect and combination potential of 0.01–1 μM rapamycin with 0–10 μM acalabrutinib on the viability of (**C**) U87MG and (**D**) LN229, as quantified through 48-h SRB assay. Combination propensity and effect were evaluated using CompuSyn software based on the Chou–Talalay algorithm for drug combinations. Combinational effects are presented as the combination index (CI), where CI < 1 indicates synergism (inner triangle), CI = 1 (on the hypotenuse) indicates an additive effect, and CI > 1 (outer triangle) indicates antagonism. * *p* < 0.05; ** *p* < 0.01; *** *p* < 0.001.

**Figure 2 pharmaceuticals-14-00876-f002:**
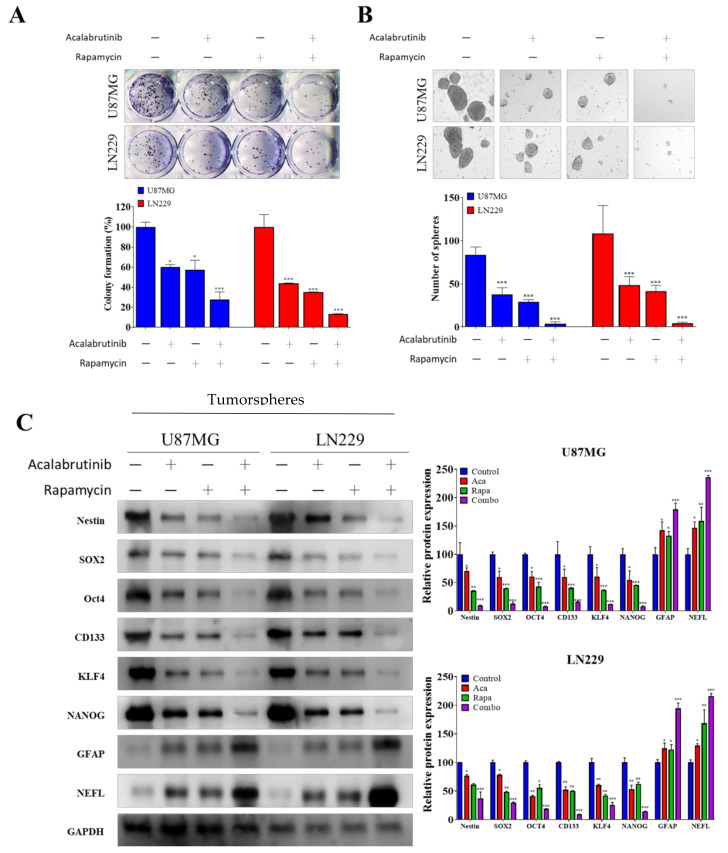
Acalabrutinib (5 µM) and rapamycin (0.01 μM) combination demonstrated enhanced efficacy in attenuating the tumorsphere-forming ability of GBM cell lines. To a greater extent than acalabrutinib or rapamycin treatment alone, the combination of acalabrutinib and rapamycin (**A**) strongly inhibited the colony-forming ability of GBM cell lines; (**B**) markedly inhibited the tumorsphere-forming ability of GBM cell lines; (**C**) Western blot assay of tumorspheres derived from U87 and LN229 cell lines revealed that the combination of acalabrutinib and rapamycin markedly attenuated the protein expression of CSCs markers Nestin, SOX2, OCT4, CD133, and KLF4, but upregulated differentiation markers GFAP and NEFL. * *p* < 0.05; ** *p* < 0.01; *** *p* < 0.001.

**Figure 3 pharmaceuticals-14-00876-f003:**
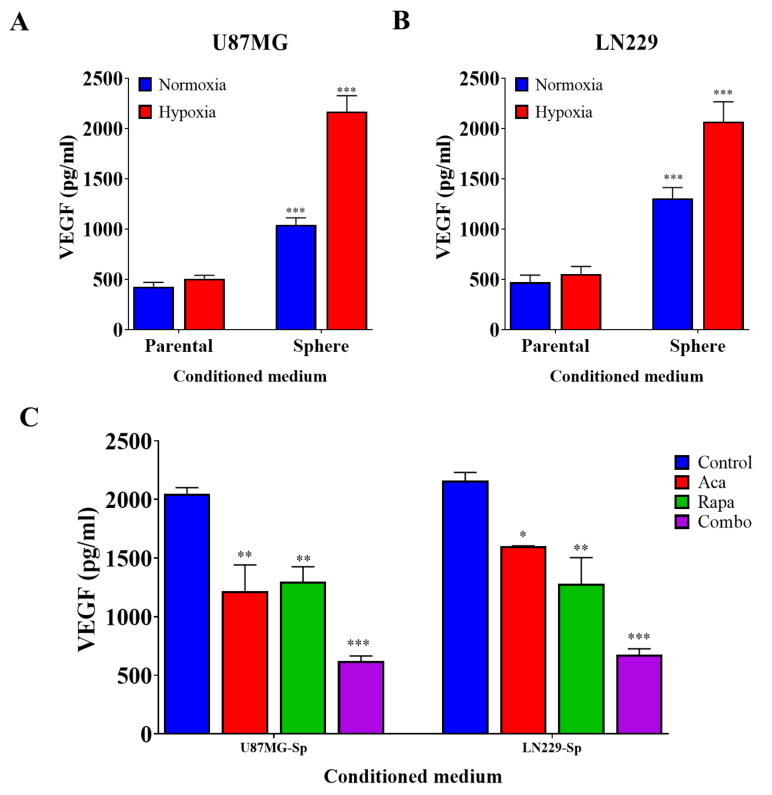
Evaluation of VEGF secretion in parental- and sphere-conditioned media through ELISA. A higher expression of VEGF was observed in the sphere-conditioned medium than in the parental-conditioned medium during hypoxia in (**A**) U87MG and (**B**) LN229 GBM cell lines. (**C**) Combination therapy with acalabrutinib and rapamycin markedly diminished VEGF expression in U87MG-sp and LN229-sp conditioned media. * *p* < 0.05; ** *p* < 0.01; *** *p* < 0.001.

**Figure 4 pharmaceuticals-14-00876-f004:**
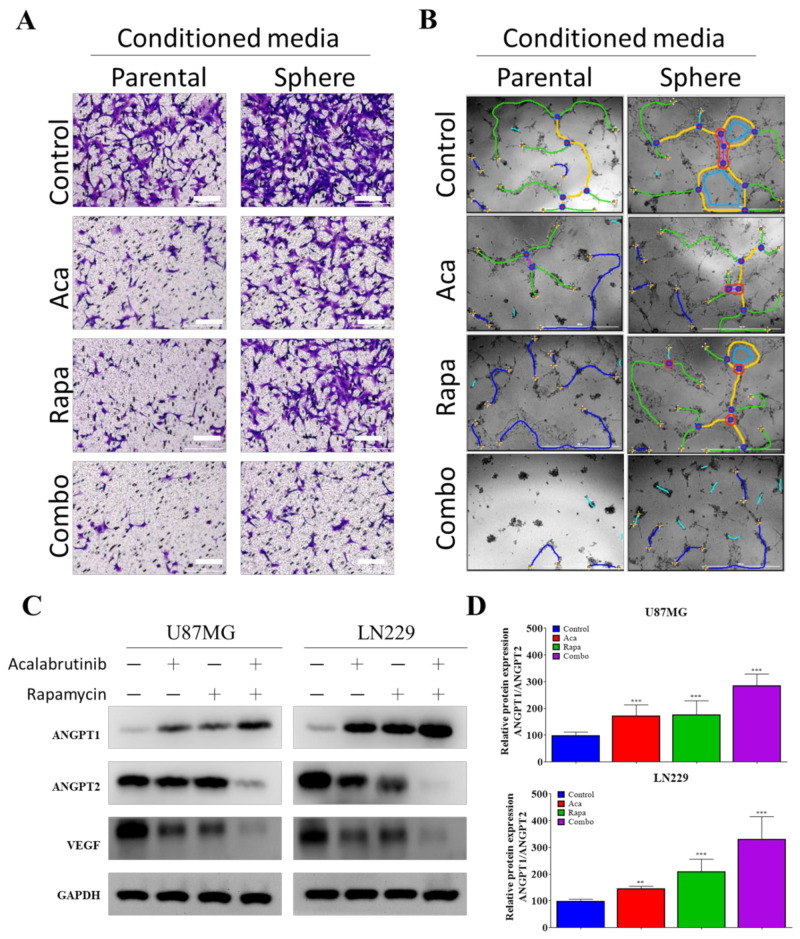
Drug-treated GBM tumorspheres had a moderate effect on the migration and tube formation of HUVECs. A combination of acalabrutinib and rapamycin (**A**) markedly inhibited the migration ability of HUVECs, (**B**) significantly inhibited the tube-formation potential of HUVECs, (**C**) and effectively increased ANGPT1/ANGPT2 ratio and diminished VEGF expression in (**D**) U87MG and LN229 cell lines. Yellow, fully formed mature tubes; red, nodes; blue, budding tube in process of forming; green, branches. ** *p* < 0.01; *** *p* < 0.001.

**Figure 5 pharmaceuticals-14-00876-f005:**
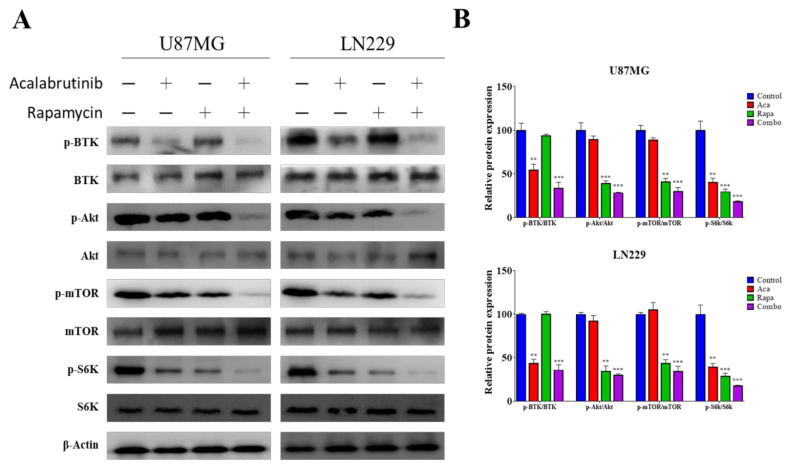
A combination of acalabrutinib and rapamycin exerted inhibitory effects on oncogenic signaling pathways in GBM cell lines. (**A**) The combination enhanced the suppression of the p-BTK, p-Akt, Akt, p-mTOR, mTOR, p-S6K and S6K protein expression levels in GBM cell lines. (**B**) Quantification of the drug combination effect showing the synergism of the drugs in the inhibition of the mTOR signaling pathway, as evidenced by enhanced phosphorylation inhibition on the protein S6K directly downstream in the mTOR pathway. ** *p* < 0.01; *** *p* < 0.001.

**Figure 6 pharmaceuticals-14-00876-f006:**
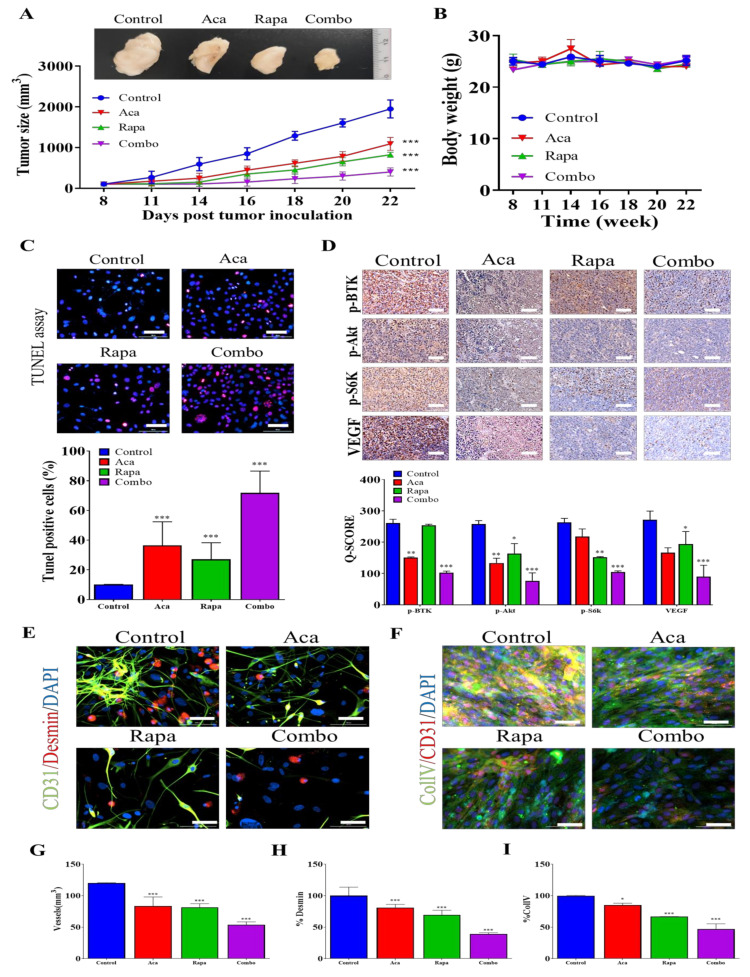
A combination of acalabrutinib and rapamycin inhibited tumorigenesis and vasculogenesis through mTOR pathway inhibition in vivo. (**A**) Graphical representation and photo images showing the effect of 20 mg/kg acalabrutinib, 4 mg/kg rapamycin, or their combination on tumor growth in U87MG-innoculated mice. (**B**) Effect of drug treatment on the weights of tumor-bearing mice. (**C**) TUNEL assay revealing the effect of drug treatment on the apoptotic rate of tumor cells. (**D**) Expression of p-BTK, p-mTOR, p-SK6 and VEGF proteins in tumor samples extracted from tumor-bearing mice. U87MG tumors were collected from mice treated with or without acalabrutinib, rapamycin, or combination therapy at the end of treatment. Sections were stained for CD31 (vessels), desmin (perivascular cells), or collagen IV (for basement membrane) and DAPI (nuclei). (**E**) Microvessel density (MVD), (**F**) perivascular cell coverage (percentage of the desmin/CD31 double-positive area in the CD31+ area), and (**G**) basement membrane coverage were greater in the dual therapy–treated tumors than in the monotherapy-treated tumors. (**H**) Representative images of CD31/desmin staining in the control, acalabrutinib, rapamycin, and combination-treated tumors on Day 21. (**I**) Representative images of CD31/Col IV staining in the control, acalabrutinib, rapamycin, and combination-treated tumors on Day 21. * *p* < 0.05; ** *p* < 0.01; *** *p* < 0.001.

**Figure 7 pharmaceuticals-14-00876-f007:**
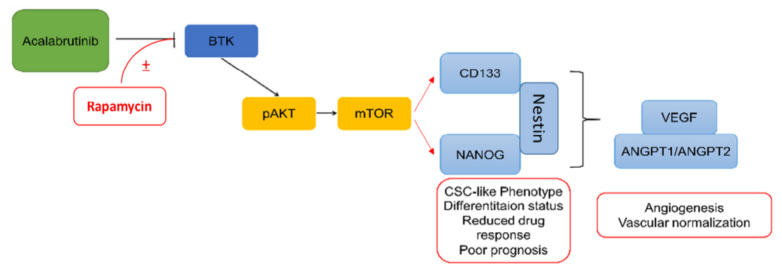
Graphical abstract depicting the synergism between acalabrutinib and rapamycin in attenuating the CSCs properties of GBM, and subsequently reduced angiogenesis and vascular normalization.

**Table 1 pharmaceuticals-14-00876-t001:** Computational analysis of HUVEC tube formation based on phase contrast.

Image Name	N0. Nodes	N0. Junctions	N0. Master Junction	N0. Master Segments	Tot. Master Segments Lenght	N0. Segments	N0. Branches	Tot. Branching Length	Tot. Segments Length	Tot. Branches Length	Branching Interval
Parental_ Control	14	5	0	1	224	4	7	875	181	694	25.857
Sphere_ Control	28	9	1	3	523	10	7	1003	489	514	69.857
Parental_ Aca	6	2	1	3	523	1	4	315	14	301	3.5
Sphere_ Aca	19	5	1	2	121	4	7	626	102	524	14.571
Parental_ Rapa	0	0	1	2	121	0	0	0	0	0	0
Sphere_ Rapa	26	7	3	4	339	6	9	948	300	648	33.333
Parental_ Combo	0	0	3	4	339	0	0	0	0	0	0
Sphere_ Combo	0	0	3	4	339	0	0	0	0	0	0

## Data Availability

The datasets used and analyzed in the current study are publicly-accessible as indicated in the manuscript.

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
