# Peer review of "Combined Treatment with Acalabrutinib and Rapamycin Inhibits Glioma Stem Cells and Promotes Vascular Normalization by Downregulating BTK/mTOR/VEGF Signaling"

_pharmaceuticals, 2021, doi:10.3390/ph14090876_

Round 1

Reviewer 1 Report

Authors here show that pharmacological benefits of combined treatment with acalabrutinib (BTK inhibitor) and rapamycin (mTOR inhibitor) in GBM by using in vitro and in vivo model. Overall, the study is well designed and manuscript is well written, and all of results and related explanations are presented well. Especially, in vivo results can provide important background to develop a way for therapeutic intervention. However, needs some improvements.

  1. Higher resolution images of tumor sphere in Figure 2B is necessary.
  2. In Figure 4C, Authors can speculate and discuss how angiopoietins are increased in combination treatment.
  3. Authors can show whether another inhibitor against mTOR and BTK also reveal synergistic efficacy in GBM cells.
  4. In Figure 7, speculation and discussion about MOA by which rapamycin regulate BTK is necessary in Discussion section.
  5. To strengthen author’s suggestion, authors can show all of tumor tissue images regarding Figure 6A.

Author Response

Dear Reviewer,

We thank the reviewer for carefully reading our manuscript and providing valuable comments. Coauthors and I very much appreciated the encouraging, critical and constructive comments on this manuscript by the reviewer. We have followed the reviewer's comments thoroughly and feel that they have further helped in strengthening the manuscript. We accordingly response the questions raised by the Reviewer as follows:

Point-by-point responses to reviewer’s comments:

Reviewer #1:

Comments and Suggestions for Authors

Journal: Pharmaceuticals

Manuscript ID: pharmaceuticals-1319121

Title: Combined treatment with acalabrutinib and rapamycin inhibits glioma stem cells and promotes vascular normalization by downregulating BTK/mTOR/VEGF signaling

Authors: Yu-Kai Su, Oluwaseun Adebayo Bamodu, I-Chang Su, Narpati Wesa Pikatan, Iat-Hang Fong, Wei-Hwa Lee, Chi-Tai Yeh, Hsiao-Yean Chiu, Chien-Min Lin *

Revision:

Authors here show that pharmacological benefits of combined treatment with acalabrutinib (BTK inhibitor) and rapamycin (mTOR inhibitor) in GBM by using in vitro and in vivo model. Overall, the study is well designed, and manuscript is well written, and all of results and related explanations are presented well. Especially, in vivo results can provide important background to develop a way for therapeutic intervention. However, needs some improvements.

R1: We thank the reviewer for this kind words and appreciate the reviewer’s interest in our work.

  1. Higher resolution images of tumor sphere in Figure 2B is necessary.

A1: We thank the reviewer for this comment. As requested by the reviewer, we have now provided higher resolution images for all our data including the one mentioned by the reviewer.

  1. In Figure 4C, Authors can speculate and discuss how angiopoietins are increased in combination treatment.

A2: We thank the reviewer for this comment. As suggested by the reviewer, we have now discussed our findings on the angiopoietins in the context of current literature in our revised discussion section. Please see Lines 515-536: 

To the best of our knowledge, this is the first study to demonstrate that BTK inhibition attenuates the angiogenesis of GBM cells and enhances vascular normalization through the elimination of GBM CSCs. Cells treated with acalabrutinib exhibited lower VEGF ex-pression and improved ANPGT1/ANPGT2 ratio. This suggests that BTK has an important role in the angiogenesis of GBM cells. Previously, everolimus, an mTOR inhibitor, was shown to synergize with ibrutinib with respect to its anticancer effects in vitro and in vivo, with BTK-mTOR inhibition leading to the inhibition of the Akt/mTOR and JAK2/STAT3 signaling pathways 22. We sought to broaden the scope of knowledge regarding the effect of this combination on the angiogenesis of GBM. Indeed, BTK signaling inhibition alone reduces the expression of proangiogenic proteins; supplementation with an mTOR inhibi-tor causes further reductions and improves the expression of antiangiogenic proteins in GBM cells. Consequently, this combination also further improves the ANGPT1/ANGPT2 ratio. This finding is of clinical significance in the light of the divergent roles of the angio-poietins; with upregulated expression of ANGPT2 associated with pathological vascular remodeling, and implicated in vascular destabilization, thus, promoting angiogenesis, tumor growth and metastasis45. Conversely, ANGPT1 has been shown to quell angiogenic signals and stabilize vessels after disruptive angiogenic processes46. More so, there is evi-dence of the prognostic significance of the ANGPT1/ANGPT2 ratio patients with primary GBM46. In addition, targeting ANGPT2 has been shown to elicit significant suppression of angiogenesis, normalize tumor-associated vessels, and inhibit metastasis47. Thus, further highlighting the potential role of BTK/mTOR inhibition as an effective antiangiogenic therapeutic strategy in patients with GBM.

Please also see Page 11, Lines 325-339:

3.4. Targeting VEGF inhibited the effects of GBM CSCs on endothelial cells

To assess the effect of the combined inhibition of BTK and mTOR on endothelial cells, we adopted an in vitro model of angiogenesis with the HUVEC cell line cocultured on matrigel with parental or sphere-conditioned media. We discovered that HUVEC grown with sphere-conditioned media exhibited increased migration (Figure 4A) and tube for-mation (Figure 4B; also see Table 1). These effects were markedly diminished when the sphere cells of U87MG were preincubated with acalabrutinib or rapamycin over the course of 48 hours. The effects were further enhanced by the simultaneous administration of both drugs on the cells (Figure 4A and 4B, also see Table 1). We confirmed that the pharmacological inhibition of BTK and mTOR further downregulated the expressions of VEGF and ANGPT2, upregulated the expression of ANGPT1, resulting in increased ANGPT1/ANGPT2 ratio in the U87MG and LN229 cell lines (Figure 4C and 4D). These results demonstrate that inhibiting VEGF expression by reducing the number of GBM tu-morspheres is a valid approach to inhibiting neoangiogenesis and enhancing vascular normalization in GBM tumors.  

  1. Authors can show whether another inhibitor against mTOR and BTK also reveal synergistic efficacy in GBM cells.

A3: We appreciate the reviewer’s suggestion. As suggested, we have now included additional data using ibrutinib and torin 2, thus showing that the synergistic efficacy in GBM is not limited to acalabrutinib and rapamycin. Please see our newly included Supplementary Figure S1 and its legend.

Supplementary Figure S1. Combined ibrutinib and torin2 exhibits synergism. Graphical representation and isobolograms of the effect and combination potential of 0.01–1 μM torin2 with 0–10 μM ibrutinib on the viability of (A) U87MG and (B) LN229, as quantified through 48-hour SRB assay. Combination propensity and effect were evaluated using CompuSyn software based on the Chou–Talalay algorithm for drug combinations. Combinational effects are presented as the combination index (CI), where CI < 1 indicates synergism (inner triangle), CI = 1 (on the hypotenuse) indicates an additive effect, and CI > 1 (outer triangle) indicates antagonism. *P < 0.05; **P < 0.01; ***P < 0.001.

Please also see Page 11, Lines 246-270:

3.1. Rapamycin and acalabrutinib effectively reduced the viability of GBM cell lines and exerted a synergistic antiproliferation effect

Previous studies have shown the effectiveness of rapamycin in inhibiting the tumor-igenesis of GBM cells. Here, we aimed to corroborate these findings and establish a basis for our working concentration in subsequent experiments. We treated GBM cell lines with various doses of rapamycin ranging from 0.01 μM up to 1 μM, with IC50 values of 0.1 μM and 0.125 μM (U87MG and LN299, respectively) (Figure 1A). Next, we used a novel, selec-tive BTK inhibitor (acalabrutinib) and demonstrated its antiproliferative effect on GBM cells. Using two GBM cell lines used in a previous experiment with rapamycin, we exam-ined the effects of acalabrutinib through SRB assay. Cells were treated with acalabrutinib at concentrations ranging between 2.5 μM and 10 μM for 48 hours. This treatment mark-edly reduced the viability of U87MG and LN299 cells, with IC50 values of at least 11 μM and 12 μM, respectively (Figure 1B). We hypothesized that BTK inhibition would enhance the effect of mTOR inhibition on GBM cell lines. Thus, we incubated U87MG and LN299 simultaneously with rapamycin and acalabrutinib for 48 hours and evaluated their effect through SRB assay. Cells exposed to rapamycin combined with acalabrutinib exhibited a considerably enhanced antiproliferative effect of rapamycin and considerably reduced vi-ability of GBM cell lines (Figures 1C and 1D). The Chou–Talalay algorithm-based Com-puSyn software was used for isobologram-aided combinatorial analysis of rapamycin with acalabrutinib, and the results demonstrated that all of the combination points within the right-angled isobologram triangle had combination index scores of <1 for all corre-sponding dose combinations, indicating synergism between the two agents. Combined ib-rutinib and torin2 also exhibits synergism on the viability of U87MG and LN229 (Supplementary Figure S1). These results highlight the potential role of therapeutically target-ing BTK and mTOR when treating patients with GBM.

  1. In Figure 7, speculation and discussion about MOA by which rapamycin regulate BTK is necessary in Discussion section.

A4: We are grateful to the reviewer for this comment. In the present study, we have shown that acalabrutinib with or without (thus, the +/- sign attached to rapamycin in the schematic abstract in Figure 7) targets and suppresses BTK expression and activity. We are not sure that rapamycin as a single agant therapy may downregulate BTK, nor were we able to demonstrate that in our present work. However, based on the reviewer’s suggestion, we do agree that it would be an interesting hypothesis to explore in our future work. Please see our updated Figure 5 and its legend, Lines 367-371:

Figure 5. A combination of acalabrutinib and rapamycin exerted inhibitory effects on oncogenic signaling pathways in GBM cell lines. (A) The combination enhanced the suppression of the p-BTK, p-Akt, Akt, p-mTOR, mTOR, p-S6K and S6K protein expression levels in GBM cell lines. (B) Quantification of the drug combination effect showing the synergism of the drugs in the inhibition of the mTOR signaling pathway, as evidenced by enhanced phosphorylation inhibition on the protein S6K directly downstream in the mTOR pathway.

Please also see our revised Results section, Lines 352-365:

3.5. Acalabrutinib inhibited BTK, Akt, and ERK signaling, enhancing rapamycin inhibition of mTOR activation

We postulated that BTK inhibition may synergize with rapamycin through its inter-action with the Akt signaling pathway, an upstream signaling axis of mTOR, because they have previously been shown to interact in B-cells 25. U87MG and LN229 cells were administered 5 µM acalabrutinib, 0.1 µM rapamycin, or both for 48 hours. Target protein expression was analyzed through western blotting. Our result demonstrated that as ex-pected, acalabrutinib inhibited the phosphorylation of BTK (p-BTK) and AKT (p-AKT) pro-teins, while rapamycin treatment had no apparent effect on the expression of Akt signaling and was only effective in inhibiting mTOR activation. However, the co-administration of acalabrutinib and rapamycin significantly suppressed the expression of p-BTK, p-AKT, p-mTOR, p-S6K, and S6K proteins (Figure 5A and 5B). This experiment revealed that acalabrutinib synergizes with rapamycin through the inhibition of BTK/Akt signaling axis, at least at the protein expression level.

  1. To strengthen author’s suggestion, authors can show all of tumor tissue images regarding Figure 6A.

A5: We thank the reviewer for this comment. In addition to the graphical representation of said data in Figure 6A, we also included representative photo images of extracted tumor tissues. We apologize that the reviewer may have missed this in our initaial submission. Please kindly see our Figure 6A and its legend, Line 413-424:

Figure 6. A combination of acalabrutinib and rapamycin inhibited tumorigenesis and vasculogenesis through mTOR pathway inhibition in vivo. (A) Graphical representation and photo images showing the effect of 20 mg/kg acalabrutinib, 4 mg/kg rapamycin, or their combination on tumor growth in U87MG-innoculated mice. (B) Effect of drug treatment on the weights of tumor-bearing mice. (C) TUNEL assay revealing the effect of drug treatment on the apoptotic rate of tumor cells. (D) Expression of p-BTK, p-mTOR, p-SK6 and VEGF proteins in tumor samples extracted from tumor-bearing mice. U87MG tumors were collected from mice treated with or without acalabrutinib, rapamycin, or combination therapy at the end of treatment. Sections were stained for CD31 (vessels), desmin (perivascular cells), or collagen IV (for basement membrane) and DAPI (nuclei). (E) Microvessel density (MVD), (F) perivascular cell coverage (percentage of the desmin/CD31 double-positive area in the CD31+ area), and (G) basement membrane coverage were greater in the dual therapy–treated tumors than in the monotherapy-treated tumors. (H) Representative images of CD31/desmin staining in the control, acalabrutinib, rapamycin, and combination-treated tumors on Day 21. (I) Representative images of CD31/Col IV staining in the control, acalabrutinib, rapamycin, and combination-treated tumors on Day 21.

Reviewer 2 Report

Su et al. present a manuscript on a highly interesting topic in glioblastoma. Several concerns arise when reading the manuscript:

-Title: oncogenic activity. Oncogenes were not studied in the manuscript. Please rephrase.

-most importantly: while the concept of glioma stem cells is still under debate, some aspects in the experimental setup/the results need to be discussed. The authors find a high expression of CD133 and Nestin in control conditions in both cell lines. Their finding ist contradictory to other, highly cited publications that have previously shown no detectable levels of Nestin in control conditions (Hong et al Int Journal of Oncology 2012 and Kurihara et al. Gene Therapy 2000). Both authors specifically studied Nestin expressions in U87 glioma cell lines without detecting any substantial signal. Further, no CD133 expression was detected in western blotting. While I understand that different antibodies might produce different results, the observed difference in CD133 expression may be explained. However, U87 was specifically described as a Nestin negative cell line by Lu et al (Int J Cancer 2011). These findings are in stark contrast with your experimental design and the results you present. Yet, you fail to discuss these observations in light of the literature.

-Further, and in connection with the previous concern, the external validity is questionable. What about other cell lines? How do cells behave that have other expression patterns of Nestin and CD133? How do expression levels/functional behaviour change when subjected to either medication/a combination of both substances?

-Discussion: anti-VEGF in Glioblastoma has clinically failed and is NOT an emerging therapy but rather declining in importance

Author Response

Reviewer #2:

Comments and Suggestions for Authors

  1. Su et al. present a manuscript on a highly interesting topic in glioblastoma. Several concerns arise when reading the manuscript:

A1: We thank the reviewer for this kind words and appreciate the reviewer’s interest in our work.

  1. -Title: oncogenic activity. Oncogenes were not studied in the manuscript. Please rephrase.

A2: We thank the reviewer for this comment. We do agree with the reviewer and as suggested, we have made some changes to the manuscript title. Please see our revised title, Lines 1-3:

Combined treatment with acalabrutinib and rapamycin inhibits glioma stem cells and promotes vascular normalization by downregulating BTK/mTOR/VEGF signaling

  1. -most importantly: while the concept of glioma stem cells is still under debate, some aspects in the experimental setup/the results need to be discussed. The authors find a high expression of CD133 and Nestin in control conditions in both cell lines. Their finding ist contradictory to other, highly cited publications that have previously shown no detectable levels of Nestin in control conditions (Hong et al Int Journal of Oncology 2012 and Kurihara et al. Gene Therapy 2000). Both authors specifically studied Nestin expressions in U87 glioma cell lines without detecting any substantial signal. Further, no CD133 expression was detected in western blotting. While I understand that different antibodies might produce different results, the observed difference in CD133 expression may be explained. However, U87 was specifically described as a Nestin negative cell line by Lu et al (Int J Cancer 2011). These findings are in stark contrast with your experimental design and the results you present. Yet, you fail to discuss these observations in light of the literature.

A3: We sincerely thank the reviewer for this very insightful comment. We are indeed cognizant of all the literature cited by the reviewer and really appreciate this interest the reviewer has shown in our findings. We apologize for the ambiguity created by our omission of vital information in our initial submission. To address the reviewer’s concern, we have now indicated the nature of the cells used in Figure 2C. Please see our revised Figure 2 and its legend, Line 298-304:

Figure 2. Acalabrutinib (5 µM) and rapamycin (0.01 µM) combination demonstrated enhanced efficacy in attenuating the tumorsphere-forming ability of GBM cell lines. To a greater extent than acalabrutinib or rapamycin treatment alone, the combination of acalabrutinib and rapamycin (A) strongly inhibited the colony-forming ability of GBM cell lines; (B) markedly inhibited the tumorsphere-forming ability of GBM cell lines; (C) Western blot assay of tumorspheres derived from U87 and LN229 cell linese revealed that the combination of acalabrutinib and rapamycin markedly attenuated the protein expression of CSCs markers Nestin, SOX2, OCT4, CD133, and KLF4, but upregulated differentiation markers GFAP and NEFL. *P < 0.05; **P < 0.01; ***P < 0.001.

Please also see our revised Results section, Page 8, Line 288-303:

3.2. Combination therapy of acalabrutinib and rapamycin considerably reduced the tumorsphere-formation potential of GBM cells

We evaluated the effects of pharmacologically blocking both BTK and mTOR pathway signaling on the maintenance of the CSCs properties of GBM cells. First, we examined the effects of 5 µM acalabrutinib and 0.1 µM rapamycin on U87MG and LN229 cell lines. The results of this experiment revealed that a single administration of each drug considerably reduced colony numbers (75% and 50% of control, respectively), and simultaneous incubation with these two drugs markedly enhanced their inhibitory effects (up to 25% compared with the control; Figure 2A). Next, we demonstrated that a small combination dose of 5 µM acalabrutinib and 0.1 µM rapamycin considerably reduced both the size and number of spheres formed by U87MG and LN229 cells by up to 49.5% and 51%, respectively (Figure 2B). Furthermore, western blot analysis of the formed GBM spheres confirmed reduction in expression of CSCs markers Nestin, SOX2, OCT4, CD133, KLF4, and NANOG but increase the expression of differentiated neuron markers GFAP and NEFL. The combination of acalabrutinib and rapamycin further enhanced or inhibited the effect of each drug on these markers (Figure 2C). These results indicated that the pharmacologic inhibition of BTK and mTOR signaling effectively abolished the CSCs-like properties of GBMs.

Consistent with our findings, we also refer the reviewer to the following reference:

(i) Hai L, Zhang C, Li T, et al. Notch1 is a prognostic factor that is distinctly activated in the classical and proneural subtype of glioblastoma and that promotes glioma cell survival via the NF-kappaB(p65) pathway. Cell Death Dis. 2018; 9(2):158.

(ii) Yao Y, Xue Y, Ma J, et al. MiR-330-mediated regulation of SH3GL2 expression enhances malignant behaviors of glioblastoma stem cells by activating ERK and PI3K/AKT signaling pathways. PLoS One. 2014; 9(4):e95060.

  1. -Further, and in connection with the previous concern, the external validity is questionable. What about other cell lines? How do cells behave that have other expression patterns of Nestin and CD133? How do expression levels/functional behaviour change when subjected to either medication/a combination of both substances?

A4: We thank the reviewer for this comment. We are not sure we fully understand what the reviewer means here. We will however try to address the question based on our perceived understanding. While the effect of Nestin and CD133 on the phenotype of GBM cells is not the principal theme of this present work, there are some published works addressing same. Please see the reference attached below:

  • Qiang l, et al showed that upregulated CD133 or Nestin enhanced the cancer stem cell-like phenotype of GBM cell lines, including U87MG, with increased ability to form tumorspheres, self-renew, and evade treatment [Qiang L, Yang Y, Ma YJ, Chen FH, Zhang LB, Liu W, Qi Q, Lu N, Tao L, Wang XT, You QD, Guo QL. Isolation and characterization of cancer stem like cells in human glioblastoma cell lines. Cancer Lett. 2009 Jun 28;279(1):13-21.]

Regarding the behavioral changes in GBM cells upon treatment with BTK inhibitor and/or MTOR inhibitor:

  • Ferrucci et al showed that Rapamycin alone dose-dependently reduces nestin expression [Ferrucci M, Biagioni F, Lenzi P, Gambardella S, Ferese R, Calierno MT, Falleni A, Grimaldi A, Frati A, Esposito V, Limatola C, Fornai F. Rapamycin promotes differentiation increasing βIII-tubulin, NeuN, and NeuroD while suppressing nestin expression in glioblastoma cells. Oncotarget. 2017 May 2;8(18):29574-29599.]
  • Chandrika et al demonstrated that torin 1 effectively reduced the expression of neural stem cell markers (Sox2, Oct4, nestin, and mushashi1) in GBM cells [Chandrika G, Natesh K, Ranade D, Chugh A, Shastry P. Mammalian target of rapamycin inhibitors, temsirolimus and torin 1, attenuate stemness-associated properties and expression of mesenchymal markers promoted by phorbol-myristate-acetate and oncostatin-M in glioblastoma cells. Tumour Biol. 2017 Mar;39(3):1010428317695921. ].
  • Wei et al showed that Ibrutinib suppressed Nestin expression in GBM cells [Wei L, Su YK, Lin CM, et al. Preclinical investigation of ibrutinib, a Bruton's kinase tyrosine (Btk) inhibitor, in suppressing glioma tumorigenesis and stem cell phenotypes. Oncotarget. 2016;7(43):69961-69975. doi:10.18632/oncotarget.11572]

  1. -Discussion: anti-VEGF in Glioblastoma has clinically failed and is NOT an emerging therapy but rather declining in importance

A5: We thank the reviewer for this rising this important point. We do agree with the reviewer that anti-VEGF as a therapeutic strategy in GBM has not lived up to its promise. This in part informs the present study, wherein we investigated the likelihood of enhancing the anti-GBM effect of VEGF inhibitors by targeting probable mechanisms by which GBM cells evade anti-VEGF killing effect, including inhibition of cancer stem cell-like phenotypes and suppression of BTK oncogenic signaling. We have revised our manuscript to reflect this sub-optimal performance of anti-VEGF agent. Please see our revised Discussion section, Lines 478-515:

Anti-angiogenic modalities are recognized as viable strategies for treating malignant tumors 34. Studies have reported the effectiveness of agents such as bevacizumab (Avastin) as a component of a combination therapy regimen in some malignancies, including colon cancer and GBM 34,35. Several theories have been proposed to clarify the clinical mechanism of antiangiogenic drugs. Some studies have revealed that some cancers  express VEGF receptors themselves, thus anti-VEGF may give them direct antitumor effects 36. Moreover, anti-angiogenic treatment have been shown to disrupt the vascular structure of tumors, depriving them of nutrition and promoting severe hypoxia 37,38. This mechanism, however, may also contribute to the perceived ineffectiveness of anti-angiogenic agent in treating GBM, as the disrupted vessels may restrict drug delivery, decreasing their clinical efficacy32. In light of this issue, researchers began to focus on another possible mechanism of anti-angiogenic agent which is that antiangiogenics may “normalize” tumor vasculature, improving its efficiency in delivering chemotherapeutic drugs and other therapeutic modalities 39,40. Several studies have shown that vascular normalization may have beneficial effect in tumors. Park et al. found that the combination of Tie2 activation and Ang2 inhibition is a potent therapeutic approach for eliciting a tumor microenvironment that favors improved chemotherapeutic drug delivery into tumors41. Recently, it has been shown that compared to VEGF-targeting alone, blocking VEGF, Ang-2, and programmed cell death protein-1 (PD-1) dramatically increased the survival of syngenic orthotopic GBM mice models42. Having said this, we do note that in a systematic review and meta-analysis of randomized clinical trials, Xiao et al concluded that while anti-VEGF agents does not improve the overall survival (OS) of patients with GBM, they do improve the progression-free survival (PFS) of these patients43

Furthermore, consistent with the findings of the present study, several studies have discussed the contribution of CSCs to tumor angiogenesis. Folkins et al. found that compared with non-CSCs, CSCs produce much higher levels of VEGF in both normoxic and hypoxic conditions and that this CSCs-mediated VEGF production leads to amplified endothelial cell migration and tube formation in vitro. When this endothelial migration and tube-formation assays were supplemented with the VEGF-blocking antibody bevacizumab, the in vitro endothelial cell behaviors were blocked. Moreover, in vivo administration of bevacizumab strongly inhibited the growth, vascularity, and hemorrhage of xenografts derived from CSCs, whereas no effects were observed on xenografts from non-CSCs 42. A VEGF-overexpression glioma model has recently provided supportive evidence for this finding by demonstrating that GBM CSCs overexpressing VEGF produce larger, more vascular, and highly hemorrhagic tumors 44. This enrichment with CSC-like cells has also been implicated in the resistance of GBM cells to anti-VEGF therapy alluded above44. Thus, the elimination of CSCs is a potential therapeutic target for improving vascular normalization in cancers.

Reviewer 3 Report

The current manuscript is an interesting study showing that BTK-mTOR inhibition using acalabrutinib and rapamycin can normalize GBM vascularization, and this may serve as a therapeutic strategy for resistant GBM. However, there are also several issues the authors should address as summarized below.

  1. Although the authors discussed about the pharmacokinetics and pharmacodynamics of acalabrutinib, they should add more content about if acalabrutinib and rapamycin can cross the blood brain barrier, which is important for GBM treatment.
  2. In section 2.9, the authors mentioned that brains with tumors were collected from U87MG-bearing mice. Is there any data/figure related to the brain samples?
  3. The figure resolution needs to be improved.
  4. Please double check the font and formatting in the main text.

Author Response

Reviewer #3:

Comments and Suggestions for Authors

The current manuscript is an interesting study showing that BTK-mTOR inhibition using acalabrutinib and rapamycin can normalize GBM vascularization, and this may serve as a therapeutic strategy for resistant GBM. However, there are also several issues the authors should address as summarized below.

We thank the reviewer for this kind words and appreciate the reviewer’s interest in our work.

  1. Although the authors discussed about the pharmacokinetics and pharmacodynamics of acalabrutinib, they should add more content about if acalabrutinib and rapamycin can cross the blood brain barrier, which is important for GBM treatment.

A1: We thank the reviewer for this suggestion. As suggested by the reviewer, we have now added some lines on the ability of acalabrutinib and rapamycin to cross the blood brain barrier. Please see our revised Discussion section, Lines 538-551:

The efficacy of any anti-GBM therapeutic would be largely dependent on the integrity of the blood-brain barrier (BBB), and that distribution of any drug across the BBB is often limited by the physico-biochemical barriers, including tight junctions, which are known to prevent paracellular transport of drugs, and efflux transporters which actively reduce the penetration and intracerebral distribution of any drug. Thus, in terms of pharmacokinetics and pharmacodynamics, earlier reports indicate that Acalabrutinib exhibits a 101 ss/F volume of distribution (Vd) to plasma, has a maximum serum concentration (Cmax) of 323 ng/mL, an area under curve (AUC) of 1111 ng h/mL, reaches Cmax in 0.75 hour (Tmax), an elimination half-life (t1/2) of 0.9 hours, and clearance rate of 159 L/h.46. Consistent with this, a recent report has shown that acalabrutinib may cross the blood-brain barrier48, while several other clinical trials to evaluate the efficacy of acalabrutinib in treating CNS tumor are in progress49,50. Moreover, a recent study has also demonstrated that ibrutinib elicits temporary disruption of the brain endothelial monolayer by inducing cytoskeletal instability and inhibiting ABC transporter51.

  1. In section 2.9, the authors mentioned that brains with tumors were collected from U87MG-bearing mice. Is there any data/figure related to the brain samples?

A3: We thank the reviewer for this question. As requested by the reviewer, we have now added the brain tumor tissue images in our revised manuscript. Please kindly see our revised Figure 6 and its legend, Lines 413-424:

Figure 6. A combination of acalabrutinib and rapamycin inhibited tumorigenesis and vasculogenesis through mTOR pathway inhibition in vivo. (A) Graphical representation and photo images showing the effect of 20 mg/kg acalabrutinib, 4 mg/kg rapamycin, or their combination on tumor growth in U87MG-innoculated mice. (B) Effect of drug treatment on the weights of tumor-bearing mice. (C) TUNEL assay revealing the effect of drug treatment on the apoptotic rate of tumor cells. (D) Expression of p-BTK, p-mTOR, p-SK6 and VEGF proteins in tumor samples extracted from tumor-bearing mice. U87MG tumors were collected from mice treated with or without acalabrutinib, rapamycin, or combination therapy at the end of treatment. Sections were stained for CD31 (vessels), desmin (perivascular cells), or collagen IV (for basement membrane) and DAPI (nuclei). (E) Microvessel density (MVD), (F) perivascular cell coverage (percentage of the desmin/CD31 double-positive area in the CD31+ area), and (G) basement membrane coverage were greater in the dual therapy–treated tumors than in the monotherapy-treated tumors. (H) Representative images of CD31/desmin staining in the control, acalabrutinib, rapamycin, and combination-treated tumors on Day 21. (I) Representative images of CD31/Col IV staining in the control, acalabrutinib, rapamycin, and combination-treated tumors on Day 21.

  1. The figure resolution needs to be improved.

A3: We thank the reviewer for this comment. As requested by the reviewer, we have now provided higher resolution images for all our data.

  1. Please double check the font and formatting in the main text.

A4: We thank the reviewer for this comment. We have now double checked the font and formatting in the main text as requested by the reviewer.

Round 2

Reviewer 2 Report

The authors have addressed my concerns sufficiently.

Reviewer 3 Report

The authors addressed all my questions and this manuscript can be accepted in present form.